# COVID-19 Vaccination in Developing Nations: Challenges and Opportunities for Innovation

Abu Baker Sheikh [1,*], Suman Pal [2], Nismat Javed [3] and Rahul Shekhar [2]

1 Department of Internal Medicine, University of New Mexico Health Sciences Center, Albuquerque, NM 87131, USA

2 Department of Internal Medicine, Division of Hospital Medicine, University of New Mexico School of Medicine, Albuquerque, NM 87131, USA; spal@salud.unm.edu (S.P.); rahul547843@gmail.com (R.S.)

3 Shifa College of Medicine, Shifa Tameer-e-Millat University, Islamabad 44000, Pakistan; nismatjaved@gmail.com

* Correspondence: absheikh@salud.unm.edu

**Abstract:** Vaccines offer a hope toward ending the global pandemic caused by SARS-CoV2. Mass vaccination of the global population offers hope to curb the spread. Developing nations, however, face monumental challenges in procurement, allocation, distribution and uptake of vaccines. Inequities in vaccine supply are already evident with resource-rich nations having secured a large chunk of the available vaccine doses for 2021. Once supplies are made available, vaccines will have to be distributed and administered to entire populations—with considerations for individual risk level, remote geography, cultural and socio-economic factors. This would require logistical and trained personnel support that can be hard to come by for resource-poor nations. Several vaccines also require ultra-cold temperatures for storage and transport and therefore the need for specialized equipment and reliable power supply which may also not be readily available. Lastly, attention will need to be paid to ensuring adequate uptake of vaccines since vaccine hesitancy has already been reported for COVID vaccines. However, existing strengths of local and regional communities can be leveraged to provide innovative solutions and mitigate some of the challenges. Regional and international cooperation can also play a big role in ensuring equity in vaccine access and vaccination.

**Keywords:** vaccine equity; hesitancy; COVAX; developing countries





## 1. Introduction

The COVID-19 pandemic has emerged as a global health crisis. Developing countries have been hit hard. Shortage of resources such as healthcare personnel, overwhelmed hospitals, overcrowding, lack of funds, 'infodemic' and a clear sense of leadership are some of the factors that worsen the situation in developing countries [1]. From the very beginning of this pandemic there was a global effort to develop an effective vaccine as a potential solution to curb the pandemic. In the last few months, a number of vaccines have been developed and approved by various regulatory authorities around the world. However, the development of effective vaccines is only the first in a long series of steps towards developing global herd immunity and halting the pandemic. Furthermore, these vaccines are associated with numerous concerns such as unexplained illnesses, clot formation, lack of trust in manufacturing companies and a rapid, unconventional mode of development. The procurement, allocation, distribution, administration and uptake of vaccines will be essential steps in the process. Developing countries are likely to face challenges at each step in the process. Since a majority of the global population resides in these nations, challenges in vaccination here need serious consideration.

## 2. Summary of Currently Available COVID-19 Vaccines

The objective of this review is to summarize the current available COVID-19 vaccines Table 1 and discuss the challenges being experienced by the developing countries in vaccination efforts against COVID-19.

**Table 1.** Summary of COVID-19 vaccines currently being used (Supplementary Table S1).

| Vaccine Name/ Manufacturer | Age (in Years) | Recommended Duration between 1st & 2nd Dose | Vaccine Efficacy * | Use in COVAX | Storage Recommendations | FDA Approved | WHO Emergency Use Listing | Affordability |
|---|---|---|---|---|---|---|---|---|
| Moderna [2] | ≥18 | 28 days | 94.0% | No [3] | −20 °C [4] | Yes [5] | No | $$$ [3] |
| Pfizer-BioNTech [6] | ≥16 | 21 days | 94.8% | Yes [3] | −80 to −60 °C [4] | Yes [5] | Yes | $$ [3] |
| Sputnik V (Gamaleya) [7] | ≥18 | 21 days | 91.6% | No [8] | 2–8 °C, −18 °C [9] | No [5] | No | $$$ [3] |
| AstraZeneca Oxford [10] | ≥18 | <6 weeks | 70.4% | Yes [3] | 2–8 °C [3] | No [5] | Yes | $ [3] |
| Sinopharm [11] | 18 to 80 | 21/28 days | 86.0% | No [3] | 2–8 °C [3] | No [5] | Yes | $$$ [3] |
| Johnson & Johnson [12] | ≥18 | Ongoing phase 3 trial (at least 47 days) | 66.0% | Yes [3] | 2–8 °C [3] | Yes [5] | Yes | $ [3] |

* vaccine efficacy: as noted in phase 3 trials, real world data may vary, COVAX—COVID-19 Vaccines Global Access—an international alliance led by the Coalition for Epidemic Preparedness Innovations (CEPI), Gavi, the Vaccine Alliance (GAVI), the World Health Organization (WHO) and the United Nations Children's Fund (UNICEF); FDA—U.S. Food and Drug Administration; Affordability: Based on lowest offered price in US dollars; <USD 10—$, USD 10–30—$$, >USD 30—$$$.

## 3. Vaccine Development

Effective vaccines for COVID-19 have been developed rapidly, indeed much more rapidly than has been seen historically for other diseases. Several factors have contributed to this. The global nature of disease and widespread coverage in mass media helped garner support and investment in research and development efforts through government and philanthropic organisations. Lessons learnt in prior pandemics led to establishment of international collaborations such as CEPI which were quickly able to support vaccine R&D. Decades of research into vaccine technology such as genome sequencing, previous mRNA vaccine candidates, adenovirus vectors, etc., have been instrumental in the successful development of COVID vaccines.

Vaccine development efforts have been a costly affair. From a global perspective, more than US $39 billion was committed for vaccine development [13]. The USA alone contributed more than US $9 billion under operation warp speed [14]. The top five companies have each received between $957 million and $2·1 billion in funding commitments, mostly from the US Government and the Coalition for Epidemic Preparedness Innovations (CEPI). The Chinese and Russian governments have invested in several vaccine candidates being developed by private companies or government enterprises.

Many developing nations lack the financial and technological resources to invest in vaccine development. They will therefore be reliant on vaccines developed in other nations and through global cooperation. Efforts at knowledge sharing through temporary waivers of intellectual property rights have been impeded by the opposition of developed nations and pharmaceutical companies [15].

With several efficacious vaccines now available, the next major challenge has been to scale up production of vaccines to meet the global need. Scaling up production is a monumental challenge with multiple components manufactured at different locations globally. Supply chain disruptions of any component in manufacturing or packaging could set back production efforts. Vaccine production capabilities are also unevenly distributed. However, partnerships have emerged between vaccine developers and manufacturers to

scale up production rapidly. For example, Oxford–AstraZeneca has projected a production of approx. 3 billion doses of vaccine in 2021 with manufacturing partnerships in various nations, most notably with Serum Institute of India, which is planned to manufacture 1 billion of these doses [16,17].

As of February 2021, 289 vaccines were in development with about 66 in different stages of trials. Ten vaccines have either been approved for use by a regulatory authority or authorized for emergency use Table 1 and Table S1 [3,5,18]. Several of these vaccines stimulate a host response against spike protein of SARS CoV2 virus. Moderna and Pfizer vaccines are mRNA vaccines, whereas the vaccines developed by Oxford–AstraZeneca and Johnson & Johnson are viral DNA vaccines using adenovirus vectors. Inactivated viral components and recombinant vectors are also being used in many, including Sinopharm and Sputnik V. Although the widely used and approved vaccines are the mRNA vaccines, inactivated vaccines such as Sinopharm from China and Covaxin from India, are being used or planned for use in many developing countries across Europe, Asia, Latin America, Middle East and Africa [19].

## 4. Procurement

The majority of the developing countries who do not have the financial and technological capabilities to develop novel vaccines may have to purchase these vaccines from open markets. A large part of the global vaccine availability has been purchased by wealthier nations pushing the developing world to the back of the queue regarding vaccine supply and delivery. For example, high-income countries represent only 16% of the world's population, but they have purchased more than half of all COVID-19 vaccine doses [20]. Fortunately, an alliance of 190 nations led by international organizations such as CEPI (Coalition for Epidemic Preparedness Innovations), GAVI (Gavi, the Vaccine Alliance), UNICEF (United Nations Children's Fund) and WHO (World Health Organization) has been set up to improve global vaccine access. The aim of the organization, COVID-19 Vaccines Global Access (COVAX), is to accelerate the development and manufacture of COVID-19 vaccines, and to guarantee fair and equitable access for every country in the world [21]. COVAX expects to secure 2 billion doses by the end of 2021 with an aim to vaccinate 20% of the population while ensuring equitable distribution such that the most vulnerable people are vaccinated first globally [20].

## 5. Administration

Once vaccine supplies are made available to developing countries, the next monumental task would be the allocation, distribution and administration of vaccines to reach entire populations. This requires not only the challenges of maintaining precise data on demographics of the population but also track vaccine uptake in real time to identify and rectify gaps. Developing nations would also need to factor in populations that have historically had poor access to healthcare, including remote, rural and migrant populations.

Several current vaccines also require ultra-cold temperatures for storage and transportation. For example, mRNA vaccines like Moderna require −20 °C for storage up to 6 months, and its stability drops to 30 days in cases of temperatures of 2 to 8 °C, and even lower at room temperature to 12 h [4]. COVID vaccines by Pfizer require even lower temperatures for storage up to 6 months (−80 to −60 °C) [4]. Other vaccines have less stringent requirements, such as the Oxford–AstraZeneca vaccine which can be stored at normal refrigeration temperatures (2 to 8 °C), and may be a viable option for vaccination in more remote locations or in regions with limited equipment or reliable power supply to maintain ultra-cold storage [22,23].

## 6. Vaccine Hesitancy

COVID vaccine hesitancy has been an emerging problem in several nations. According to recent estimates from a survey, some developing nations such as India reported a higher

willingness for vaccination but other countries such as Serbia, Croatia, France, Lebanon and Paraguay were on the lower end of the acceptability spectrum [3].

There are a few issues contributing to COVID-19 vaccine hesitancy. In many countries, the vaccines had been either bought or developed at a very fast pace raising concerns that the trials were rushed and regulatory standards relaxed [24]. Another concern is that the pandemic brought on the use of the very first mRNA vaccine. The novelty of the approach alone has sparked some hesitancy [24]. Many people lack trust in the manufacturing companies [25,26]. There is also an ongoing disinformation campaign against COVID-19 vaccines on multiple social media platforms [27]. For developing countries, concerns about composition of the vaccine and its acceptability for religious and ethnic groups are widely present [28,29].

The safety of vaccines is also being brought to question. The important role of Oxford's AstraZeneca vaccine was effectively undermined when scientists began to report cases of thrombosis [30]. The trial for the Adenovirus 26 vector based vaccine, being manufactured by Johnson & Johnson was similarly put on hold when one of the participants developed an unexplained illness [31]. This placed underdeveloped countries at a grave risk because the most viable and cost-effective option was now no longer available.

According to a recent survey, certain factors were presented as predictors of accepting COVID-19 vaccination. The most important predictor was the participant's vaccination against influenza. The acceptance was more in physicians than nurses, doctors working as part of internal medicine than other teams and doctors working in COVID-19 units ($p < 0.01$) [29]. Participants whose means of earning had been severely impacted by the crisis were more accepting towards the idea of a vaccination. Interestingly, age was not found to be an important factor [29].

Political factors are also responsible for vaccine hesitancy among the masses. A few prior incidents such as the French government overestimating the need for vaccines and voting bias have created mistrust among the people and thus breaking down a strong communication network [32]. In rural sectors, there is limited epidemiological data available in primary healthcare systems. Although there have been previous attempts at vaccination by creating a logic network, the question of villages not recognized on census still remains [33].

## 7. Leveraging Existing Strengths

While many developing nations face monumental challenges in their vaccination efforts, it is essential to look at existing strengths in these regions and leverage them to help vaccination efforts. Many developing nations have had recent or ongoing vaccination campaigns against communicable diseases. Eradication of wild polio in Africa is a success story of regional cooperation, infrastructure and expertise building. These resources —surveillance networks, trained personnel, and operation centers—can be utilized in supporting COVID-19 vaccination efforts [34]. Ongoing polio vaccination campaigns in Pakistan means that healthcare workers are already present in and familiar with local communities, which could be utilized not just for vaccine delivery but also to boost uptake by targeted messaging from trusted sources. India's universal immunization program, the largest public health program in the world, and vaccine manufacturing facilities such as Serum Institute of India, also the largest in the world, are robust and readily available resources that will pivot to support the vaccination effort in the nation [35].

Strategies in other countries can also help reform the response to COVID-19 vaccination. This includes attempts to eradicate Hepatitis B in sub-Saharan Africa [36]. Although intrauterine transmission of COVID-19 seems unlikely, it has led to detrimental outcomes in neonates [37]. Antenatal discussions can play a role in this regard. By engaging pregnant women to promote vaccination as Africa did for Hepatitis B, more awareness can be spread. However, it should be explained that vaccination might not prevent neonatal infection and can reduce infection rates in pregnant women leading to less complications in routine antenatal care [36,37].

Many developing nations have also learnt important lessons from missteps made in other public health interventions. Recognizing these and adapting COVID vaccination efforts accordingly can help avoid costly errors. For example, a disinformation effort in Pakistan had setbacks in the nations' effort to eradicate wild polio. Learning from this, the nation is proactively preparing to counter misinformation against COVID vaccine [38].

Another example can be found in Nigeria's efforts towards polio eradication. During the initial phases of the public health intervention, political leaders boycotted the movement and were of the idea that the vaccines had antifertility chemicals, HIV and carcinogenic agents [39]. This created a sense of mistrust with the public and many parents refused to get their children vaccinated. At the time, religious scholars played an important role. They encouraged people to undergo vaccination because it did not violate any principles related to their religion, culture or basic human rights. In fact, they elaborated on the negative impact that such a boycott produces on the same religion globally. The intervention on behalf of these leaders is one of the ways in which the nation is tackling its polio crisis and should serve as inspiration for other nations to follow [39].

Political stability continues to be a cornerstone of effective eradication programs. This lesson was learnt from the historical example of smallpox eradication in Somalia [39]. The years 1977 to 1979 were probably known as the deadly years because of the enormous spread of smallpox. What undermined the efforts of programs at that time was the Somalian–Ethiopian war among other factors. It is important to note how a third more influential neutral party, in this case, WHO, had to intervene and tackle the situation by providing resources [40]. Figure 1 summarizes the proposed public health strategy for COVID-19 vaccination.

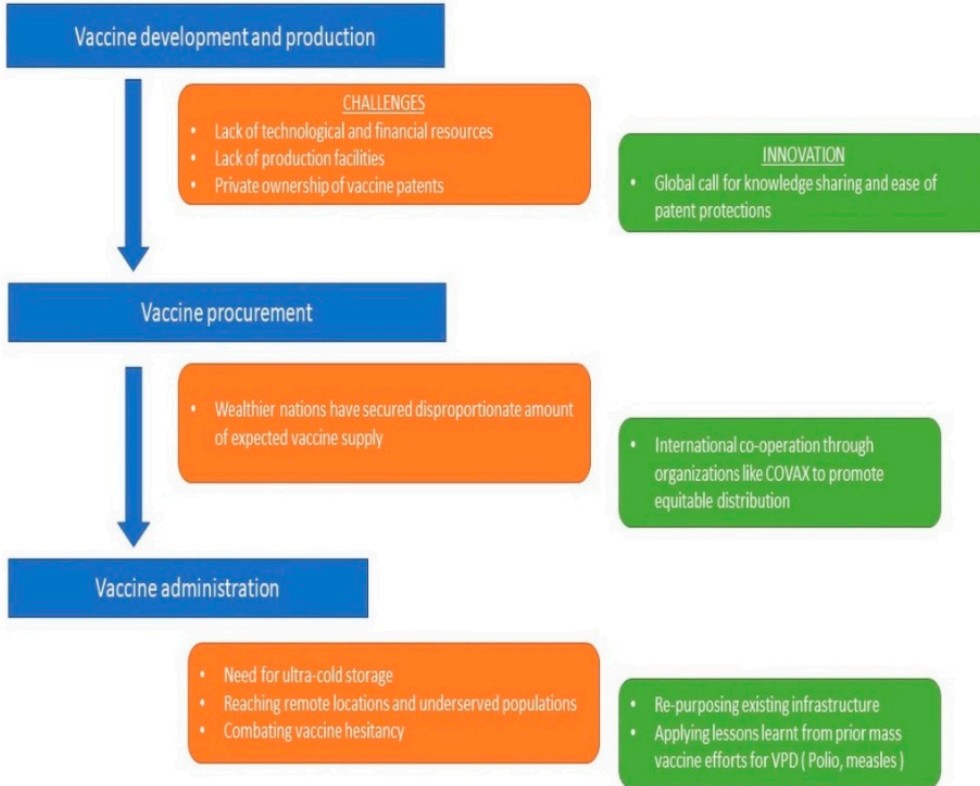

**Figure 1.** Vaccination as a public health strategy in developing nations: facing challenges with innovations and existing community strengths.

## 8. Conclusions

Although many global initiatives have taken up the responsibility of ensuring vaccine equity, there is a need to maintain sustainability of these initiatives for future disasters. Apart from acquiring vaccines, it is important to ensure appropriate administration of vaccines by involving religious, cultural and social representatives. Leveraging existing strengths can help mitigate some of the challenges facing developing nations in vaccinating populations and bringing the world closer to the end of this pandemic.

**Supplementary Materials:** The following are available online at https://www.mdpi.com/article/10.3390/idr13020041/s1.

**Author Contributions:** Conceptualization, A.B.S., S.P., N.J. and R.S.; Methodology, A.B.S., S.P. and N.J.; Software N/A; Validation, S.P. and R.S.; Formal Analysis, N/A; Investigation, N/A; Resources, R.S.; Data Curation, N/A; Writing—Original Draft Preparation, A.B.S. and N.J.; Writing—Review and Editing, S.P. and R.S.; Visualization, N/A; Supervision, R.S.; Project Administration, N/A; Funding Acquisition, N/A. All authors have read and agreed to the published version of the manuscript.

**Funding:** This research received no external funding.

**Institutional Review Board Statement:** Not applicable.

**Informed Consent Statement:** Not applicable.

**Data Availability Statement:** Not applicable.

**Conflicts of Interest:** The authors declare no conflict of interest.

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
