# Peer review of "COVID-19 Vaccination in Developing Nations: Challenges and Opportunities for Innovation"

_2036-7449, doi:10.3390/idr13020041_

Round 1

Reviewer 1 Report

The authors selected an important and relevant topic on COVID-19 vaccination and wrote the review article very well. The authors provide insightful information and novel strategies that worth consideration by developing countries in their COVID-19 vaccination efforts.

It would be better if the authors can explain more about the “COVAX” or efforts like this that help COVID-19 vaccination in developing countries.

In addition, the authors need to pay attention to typos in lines 107 and 109 on page 4.

Author Response

Dear Respected Reviewer,

Thank you for your valuable feedback that has helped us in improving our manuscript.

Reviewer Comment 1 - It would be better if the authors can explain more about the “COVAX” or efforts like this that help COVID-19 vaccination in developing countries.

Response- We agree with the reviewer and have added this elaboration beginning from line 95 “Fortunately, an alliance of 190 nations led by international organizations…” on Page 4

Reviewer Comment 2- The authors need to pay attention to typos in lines 107 and 109 on page 4.

Response – We thank the reviewer for the suggestion. The typographical errors in lines 107 and 109 have been corrected.

Reviewer 2 Report

Review report MDPI

Title: covid-19 vaccination in developing nations: challenges and opportunities for innovations

Summary:

The review manuscript summarizes information regarding currently available vaccines against covid-19 pandemic. It includes paragraphs about vaccine development, procurement, administration, vaccine hesitancy and a final paragraph with comments from the authors to improve vaccination procedures.

General comments:

Currently there are many papers available every day about SARS-cov2. The information comes from news, radio, social media and at some point, all the information becomes repetitive, but of course necessary for everyone to know where we are at this current pandemic. Personally, and with all respect to authors, I find this review repetitive and doesn’t really include information that we already known about vaccines and covid-19. At first, when I had the task to review a manuscript about vaccines against SARS-COV2 I saw a good opportunity to update and refresh my knowledge about currently available vaccines, but I rather found long paragraphs that pinpoint at aspects that everyone knows by now such as: scale up vaccine production, developing countries don’t have the financial and technological capabilities to develop novel vaccines. Again, with all respect, I don’t think that the scientific community really needs a review to understand this.

There are several aspects about vaccines that could be reviewed and that could be of great help for the scientific community such as: cellular and humoral immune response, measurement of vaccine efficacy, long term immunity, virus ‘evolution’ and the rise of new viral strains, how new variants could hamper vaccination procedures and also diagnostic. But I guess these modifications will change the entire manuscript and its scope.

Finally, as this is a review, I would suggest the authors to include more graphs or diagrams to explain data more rapidly and easily.

Specific comments:

Table 1: RCT is not defined

Table 1: Age in y mean years?

Table 1: Mean vaccine efficacy: Could the authors explain more in the introduction how is the efficiency usually measured?

Table 1: What COVAX stand for. Please define or add a footnote at least. As it contains a yes/no nomenclature I assume it means approval from WHO?

Table 1: Public concern? I would suggest to the authors to add this information in the introduction and remove it from the table as it enlarge the presented information and doesn’t add any solid or scientific information. For example: Lack of trust: How could the authors clearly define the ‘public’ that do not trust in companies?

Table 1: Define FDA either as footnote or in the introduction.

Table 1: Affordability. I see symbols here that represent $$$:higher price and $ lower price?? Is this correct? If so, the authors should make it clear what they mean with these symbols.

Table 1: Is Gamaleya included in this table?

Table 1: Perhaps including a column that mention authorisation by WHO or any other regulatory body would be useful.

Line 79 – 80. The authors mention 10 vaccines and in table 1 they have just 6. They even mention 289 vaccines in development and 66 in different stages of trials. Would be good to have at least the 66 ones mentioned either in a new table or as supplementary material.

Line 106 to 113. Good point about vaccine storage. I suggest include this information in Table 1 as this is relevant and should be considered rapidly by people using vaccines.  

Author Response

Dear Respected Reviewer,

Thank you for your valuable feedback that has helped us in improving our manuscript.

Reviewer Comment 1: Table 1: RCT is not defined, age in y mean years?

Response: These columns from the table have been amended; the column for study design has been deleted and age (years) has been defined in headings for Table 1.

Reviewer Comment 2: Table 1: Mean vaccine efficacy: Could the authors explain more in the introduction how is the efficiency usually measured?

Response: This efficacy is based on preliminary reports from phase 3 trials and does not represent real-world data that might vary. This has been explained as a footnote for table 1.

Reviewer Comment 3: Table 1: What COVAX stand for. Please define or add a footnote at least. As it contains a yes/no nomenclature I assume it means approval from WHO?

Response: COVAX or COVID-19 Vaccines Global Access is an international alliance led by Coalition for Epidemic Preparedness Innovations (CEPI), Gavi, the Vaccine Alliance (GAVI), World Health Organization (WHO) and United Nations Children’s Fund (UNICEF). The yes/no nomenclature represents participation in the alliance. This has been explained as a footnote in Table 1.

Reviewer Comment 4: Table 1: Public concern? I would suggest to the authors to add this information in the introduction and remove it from the table as it enlarge the presented information and doesn’t add any solid or scientific information. For example: Lack of trust: How could the authors clearly define the
‘public’ that do not trust in companies?

Response: The column has been deleted from the table. A few of the issues causing concern among the masses have been explained on Page 2,” Furthermore, these vaccines are associated with numerous
concerns..”. A further explanation has been provided in discussion under vaccine hesitancy on Page 4, Paragraph 3 as highlighted information and lines “Many people lack..”. “The trial for the..”.

Reviewer Comment 5: Table 1: Define FDA either as footnote or in the introduction. Affordability. I see symbols here that represent $$$: higher price and $ lower price?? Is this correct? If so, the authors should make it clear what they mean with these symbols.

Response: FDA has been defined in the footnote as US Food & Drug Administration. Additionally, the price ranges for vaccine doses in US dollars have also been described in the footnote as “Based on lowest offered price in US dollars; <USD 10 - $, USD 10-30 - $$, >USD 30 - $$$”

Reviewer Comment 6: Table 1: Is Gamaleya included in this table?

Response: Gamaleya is the manufacturer of a vaccine named Sputnik V. This has been added to the table next to Sputnik V.

Reviewer Comment 7: Table 1: Perhaps including a column that mention authorisation by WHO or any other regulatory body would be useful.

Response: This column has been added in Table 1 and includes authorization status by WHO as well as FDA.

Reviewer Comment 8: Line 79 – 80. The authors mention 10 vaccines and in table 1 they have just 6. They even mention 289 vaccines in development and 66 in different stages of trials. Would be good to have at least the 66 ones mentioned either in a new table or as supplementary material.

Response: The six vaccines mentioned in the article were in common use in different parts of the world at the time of writing the review. Four additional vaccines which have also been in use/ slated to be in use soon are added to Supplementary Table 1. The data on the remaining vaccine candidates is limited
and there they have therefore not been included in the table.

Reviewer Comment 9: Line 106 to 113. Good point about vaccine storage. I suggest include this information in Table 1 as this is relevant and should be considered rapidly by people using vaccines.

Response: Vaccine storage requirements have been included in table 1.

General comment by Reviewer: Currently there are many papers available every day about SARS-cov2. The information comes from news, radio, social media and at some point, all the information becomes repetitive, but of course necessary for everyone to know where we are at this current pandemic. Personally, and with all respect to authors, I find this review repetitive and doesn’t really include information that we already know about vaccines and covid-19. At first, when I had the task to review a manuscript about vaccines against SARS-COV2 I saw a good opportunity to update and refresh my knowledge about currently available vaccines, but I rather found long paragraphs that pinpoint at aspects that everyone knows by now such as: scale up vaccine production, developing countries don’t have the financial and technological capabilities to develop novel vaccines. Again, with all respect, I don’t think that the scientific community really needs a review to understand this. There are several aspects about vaccines that could be reviewed and that could be of great help for the scientific community such as: cellular and humoral immune response, measurement of vaccine efficacy, long term immunity, virus ‘evolution’ and the rise of new viral strains, how new variants could hamper vaccination procedures and also diagnostic. But I guess these modifications will change the entire manuscript and its scope. Finally, as this is a review, I would suggest the authors to include more graphs or diagrams to explain data more rapidly and easily.

Response: We thank the reviewer for their consideration. We appreciate that the current COVID-19 pandemic has seen a rapid rise in the volume of published literature and that reviews of topics can seem repetitive. We also acknowledge that in the rapidly evolving world of covid vaccine development, a comprehensive overview of currently existing knowledge on vaccines may be useful. However, our intent with this review is not to recapitulate the existing knowledge on vaccines ( which is also likely to become obsolete by the time of publication) but rather to focus on vaccination as a process and highlight the
challenges that developing nations may face in achieving the goal of vaccinating their populations. We agree with the reviewer that it may seem self-evident that developing nations lack resources, however, this needs to be highlighted and mitigated on a global scale. Indeed, since we wrote this review, the catastrophic rise in cases in India coupled with a very low rate of vaccination in the country
(which was slated to be the manufacturing hub for vaccine supply for much of the developing world) shows just how relevant it is to acknowledge and rapidly address the challenges that developing nations face. Therefore, at this time, we would like to keep the scope of our review limited to this issue and not
expand it as suggested.

We accept the reviewer’s suggestion to include more visual data and have included Figure 1 which we hope summarizes the salient points of our review.

Round 2

Reviewer 2 Report

The authors included most of the requested information

I would recommend double check references as most of them are underlined and it is very difficult to follow them correctly